# The relationship between lipoprotein A and other lipids with prostate cancer risk: A multivariable Mendelian randomisation study

**Anna Ioannidou**[1], **Eleanor L. Watts**[2], **Aurora Perez-Cornago**[2], **Elizabeth A. Platz**[3,4,5], **Ian G. Mills**[6,7,8], **Timothy J. Key**[2], **Ruth C. Travis**[2], **The PRACTICAL consortium, CRUK, BPC3, CAPS, PEGASUS**[¶], **Konstantinos K. Tsilidis**[1,9☯], **Verena Zuber**[1☯*]

1 Department of Epidemiology and Biostatistics, School of Public Health, Imperial College London, London, United Kingdom, 2 Cancer Epidemiology Unit, Nuffield Department of Population Health, University of Oxford, Oxford, United Kingdom, 3 Department of Epidemiology, Johns Hopkins Bloomberg School of Public Health, Baltimore, Maryland, United States of America, 4 Sidney Kimmel Comprehensive Cancer Center at Johns Hopkins, Baltimore, Maryland, United States of America, 5 Department of Urology and the James Buchanan Brady Urological Institute, Johns Hopkins University School of Medicine, Baltimore, Maryland, United States of America, 6 Nuffield Department of Surgical Sciences, University of Oxford, Oxford, United Kingdom, 7 Patrick G Johnston Centre for Cancer Research (PGJCCR), Queen's University Belfast, Belfast, United Kingdom, 8 Centre for Cancer Biomarkers (CCBIO), University of Bergen, Bergen, Norway, 9 Department of Hygiene and Epidemiology, University of Ioannina School of Medicine, Ioannina, Greece

☯ These authors contributed equally to this work.
¶ Members from the PRACTICAL Consortium, CRUK, BPC3, CAPS, and PEGASUS are provided in the Acknowledgments section.
* v.zuber@imperial.ac.uk

**Data Availability Statement:** Our study was conducted in a two-sample, summary-level Mendelian randomization design. The exposure

## Abstract

### Background

Numerous epidemiological studies have investigated the role of blood lipids in prostate cancer (PCa) risk, though findings remain inconclusive to date. The ongoing research has mainly involved observational studies, which are often prone to confounding. This study aimed to identify the relationship between genetically predicted blood lipid concentrations and PCa.

### Methods and findings

Data for low-density lipoprotein (LDL) cholesterol, high-density lipoprotein (HDL) cholesterol, triglycerides (TG), apolipoprotein A (apoA) and B (apoB), lipoprotein A (Lp(a)), and PCa were acquired from genome-wide association studies in UK Biobank and the PRACTICAL consortium, respectively. We used a two-sample summary-level Mendelian randomisation (MR) approach with both univariable and multivariable (MVMR) models and utilised a variety of robust methods and sensitivity analyses to assess the possibility of MR assumptions violation. No association was observed between genetically predicted concentrations of HDL, TG, apoA and apoB, and PCa risk. Genetically predicted LDL concentration was positively associated with total PCa in the univariable analysis, but adjustment for HDL, TG, and Lp(a) led to a null association. Genetically predicted concentration of Lp(a) was

data on genetic associations with lipoproteins was provided by the Nealelab (http://www.nealelab.is/uk-biobank). The outcome data on genetic associations with total prostate cancer is available for download from the PRACTICAL consortium (http://practical.icr.ac.uk/), genetic association data on advanced and early-age onset prostate cancer is restricted access, but available from the PRACTICAL consortium upon application (contact: PRACTICAL@icr.ac.uk).

**Funding:** KKT was supported by Cancer Research UK (C18281/A29019). ELW was supported by the NDPH Early Career Research Fellowship. APC is supported by a Cancer Research UK Population Research Fellowship (C60192/A28516) and by the World Cancer Research Fund (WCRF UK), as part of the Word Cancer Research Fund International grant programme (2019/1953). The funders had no role in study design, data collection and analysis, decision to publish, or preparation of the manuscript.

**Competing interests:** I have read the journal's policy and the authors of this manuscript have the following competing interests: VZ is a paid statistical consultant on PLOS Medicine's statistical board.

**Abbreviations:** ALT, alanine aminotransferase; apoA, apolipoprotein A; apoB, apolipoprotein B; AST, aspartate aminotransferase; BMI, body mass index; GS, Gleason score; GWA, genome-wide association; HDL, high-density lipoprotein; IV, instrumental variable; IVW, inverse variance weighting; LD, linkage disequilibrium; LDL, low-density lipoprotein; Lp(a), lipoprotein A; MDCS, Malmö Diet and Cancer Study; MR, Mendelian randomization; MVMR, multivariable MR; OR, odds ratio; PCa, prostate cancer; PSA, prostate-specific antigen; SD, standard deviation; SNP, single nucleotide polymorphism; TG, triglyceride.

associated with higher total PCa risk in the univariable ($OR_{weighted\ median}$ per standard deviation (SD) = 1.091; 95% CI 1.028 to 1.157; $P$ = 0.004) and MVMR analyses after adjustment for the other lipid traits ($OR_{IVW}$ per SD = 1.068; 95% CI 1.005 to 1.134; $P$ = 0.034). Genetically predicted Lp(a) was also associated with advanced (MVMR $OR_{IVW}$ per SD = 1.078; 95% CI 0.999 to 1.163; $P$ = 0.055) and early age onset PCa (MVMR $OR_{IVW}$ per SD = 1.150; 95% CI 1.015,1.303; $P$ = 0.028). Although multiple estimation methods were utilised to minimise the effect of pleiotropy, the presence of any unmeasured pleiotropy cannot be excluded and may limit our findings.

## Conclusions

We observed that genetically predicted Lp(a) concentrations were associated with an increased PCa risk. Future studies are required to understand the underlying biological pathways of this finding, as it may inform PCa prevention through Lp(a)-lowering strategies.

## Author summary

### Why was this study done?

- Prostate cancer (PCa) is geographically and clinically very heterogeneous, and, as a result, its risk factors may differ according to disease aggressiveness.

- The established PCa risk factors are mainly non-modifiable, which challenge PCa prevention efforts.

- Previous observational research has identified associations between blood lipids and PCa, though results remain inconclusive.

- The aim of this study was to identify evidence for any association between several blood lipids (i.e., LDL, HDL, TG, apoA, apoB, and Lp(a)) and total, advanced, as well as early age onset PCa.

### What did the researchers do and find?

- The researchers used genetic variants that are known to be associated with each of the blood lipids, to test whether they were associated with any of the 3 PCa outcomes.

- This Mendelian randomisation (MR) analysis can reduce the existence of confounding factors and reverse causation, given that genetic variants are randomly allocated and independently assorted during meiosis. MR provides complementary evidence to observational research.

- This study provided evidence for a positive association between genetically predicted lipoprotein A (Lp(a)) concentrations, but not with other lipids, and risk of total, advanced, and early age onset PCa.

## What do these findings mean?

- Elevated Lp(a) could play a potentially important role in increasing the risk of PCa.

- It remains, however, unclear whether Lp(a) is the causal factor, given that its pathophysiological mechanisms have not been well studied.

- These findings provide rationale for further Lp(a) research to understand its functionality and role in PCa, which could lead to repurposing lipid drugs for high-risk individuals that target Lp(a) directly and study their effectiveness against PCa.

## Introduction

Prostate cancer (PCa) is one of the most frequently diagnosed cancers in men [1], with 1,276,106 incident cases reported globally during 2018 [2]. There is high geographical heterogeneity of PCa incidence, which is reflected in a 40-fold difference in the age-adjusted incidence rates across the globe [3]. Several studies have argued that this could be attributed to the increased number of diagnoses in countries where the prostate-specific antigen (PSA) screening is prevalent. Nevertheless, the basis for this heterogeneity remains poorly understood [1].

Given that PCa is also clinically heterogeneous, risk factors identified to date differ by disease aggressiveness [4]. In particular, established risk factors for total PCa are mainly nonmodifiable, including older age, African descent, and genetics [5], whereas some potential risk factors for aggressive PCa include smoking, obesity [6], lower vitamin D, and higher blood lipid levels [4], which are modifiable. Lipid-lowering therapies are cheap and well established for lowering cardiovascular risk. Yet, there is no conclusive evidence that repurposed lipid-lowering drugs are effective for the prevention of PCa. It is therefore important to determine whether blood lipids increase PCa risk, especially lethal disease [7]. A meta-analysis of 14 prospective studies published in 2015 [8] did not observe significant associations between triglyceride (TG), high-density lipoprotein (HDL), or low-density lipoprotein (LDL) concentrations and risk of total or high-grade PCa, but high between-study heterogeneity was evident for most associations. Two meta-analyses have examined the role of statin use in PCa risk, and both observed inverse associations of statins and advanced PCa risk [9,10]. Nonetheless, whether these associations can be attributed to lower cholesterol itself or some other mechanism is unknown.

Observational studies may suffer from unobserved confounding and reverse causation [11], which could explain inconsistent findings among studies. Mendelian randomisation (MR) uses genetic variants as proxies for the exposures of interest and, if carefully conducted, can complement observational research [12] and support triangulation of evidence. That is because genetic variants are randomly allocated to offspring by parents and independently assorted during meiosis, which minimise issues with reverse causation and confounding [11,13]. In addition, most studies on lipids and PCa measure lipid levels only once, which can lead to measurement error in the findings, whereas genetically predicted lipid levels capture lifelong expected levels. Previous MR research is limited to 2 studies that examined the role of HDL, LDL, and TG in PCa risk overall and by disease stage and grade, and both reported null associations [14,15]. However, neither study adjusted for multiple lipid traits, which may have limited their findings, given that different lipids are correlated and pleiotropic [16]. In this paper, we aim to identify whether genetically predicted lipid traits are associated with overall PCa risk

and, in particular, advanced and early age onset disease. We incorporated a summary-level two-sample univariable and multivariable MR (MVMR) framework to adjust for pleiotropic lipid effects and examined the role of HDL, LDL, and TG, as well as additional lipid traits that have been underexamined to date, such as lipoprotein A (Lp(a)), apolipoprotein A (apoA), and apolipoprotein B (apoB).

## Methods

### Study populations

Our study design followed a summary-level two-sample MR framework and thereby made use of lipids and PCa data from 2 different sources.

### Blood lipids data

Genome-wide association (GWA) data for HDL, LDL, TG, Lp(a), apoA, and apoB were available from UK Biobank, with information on over 13.7 million single nucleotide polymorphisms (SNPs) and downloaded from the Neale lab [17]. Model adjustments in this UK Biobank GWAS from the Neale lab included age, age^2, sex (as inferred by genotype), interaction terms for age*sex and for age^2*sex, and the first 20 principal components. All measured serum biomarkers were approximately normally distributed except Lp(a), which was positively skewed. For consistency purposes, inverse rank-normalised data were used for all biomarkers. When performing an MR analysis, it is important that the exposure can be strongly predicted by genetic variants. Heritability estimates for each of the lipid traits were reported by Sinnott-Armstrong and colleagues [18], were based on the HESS algorithm [19], and were reported as follows: HDL 36%, LDL 29%, TG 29%, Lp(a) 24%, apoA 31%, and apoB 32%, indicating strong genetic regulation of all lipid traits considered as exposures (S1 Table). For the purpose of this research and to match with the PCa GWAS, only European ancestry male participants were included ($N = 167,020$).

### PCa data

Summary association statistics for PCa risk were acquired from the PRACTICAL consortium and are based on Schumacher and colleagues [20]. More information on the included study designs (cohort and case–control studies) and participant selection can be found in the original GWAS and in S2 Table. The genotyping was performed using a custom array, namely the OncoArray. For our analysis, we used total, advanced (metastatic or Gleason score (GS) $> = 8$ or PSA $> 100$ ng/mL or PCa death) and early age onset (PCa age $< = 55$) PCa. Study participants for total PCa make up to a total of 79,166 cases and 61,106 controls, advanced PCa cases include 15,167 participants and 58,308 controls, whereas early age onset PCa includes 6,988 cases and 44,256 controls. All participants were of European ancestry.

### Assumptions

The following assumptions were made for all MR analyses and are described in combination for both the univariable and MVMR approaches [21].

1. **Relevance:** Genetic variants are associated with the exposure of interest in the case of univariable MR, whereas for MVMR, they are associated with at least one of the exposures.

2. **Exchangeability:** Genetic variants are independent of all confounders of the exposure–outcome association for the univariable MR, whereas in the MVMR, variants are independent of all confounders of each of the exposure–outcome associations.

3. **Exclusion restriction:** Genetic variants are independent of the outcome given the exposure/s and all the confounders.

## Main MR analyses

All MR analyses were performed in R version 4.0.0. Due to the availability of exposure (blood lipids) and outcome (PCa) data from 2 different sources, we used a two-sample MR study design. In the univariable MR, SNPs that satisfied genome-wide significance ($P < 5 \times 10^{-8}$) were selected for each trait. As we combined summary-level data from 2 sources, we removed inconsistencies in cases where neither the effect nor the noneffect alleles matched for a single SNP between the 2 datasets. Such cases can occur for a biallelic SNP when one dataset reports the effect of an SNP using a pair of alleles on the positive strand, whereas the other dataset reports the pair for the same SNP on the negative strand [22]. Upon removing these inconsistencies, we harmonised the data so that the exposure and outcome datasets would have the same effect allele. We used the TwoSampleMR package version 0.5.4 to clump the data using a threshold of $r2 < 0.001$ to identify and remove any SNPs in linkage disequilibrium (LD). All SNPs left after clumping were considered as the instrumental variables (IVs). We firstly ran the univariable analysis on all blood lipids for both total and advanced PCa. Following peer review comments, this analysis was also performed for all blood lipids and early age onset PCa. For the main estimation methods of the univariable analyses, we performed the inverse variance weighting (IVW) [23,24] and weighed median [11] approaches, and we additionally applied the MR-Egger [25] approach, using the MendelianRandomization package version 0.4.2.

To adjust for different lipid traits in our models, we performed an MVMR analysis. We chose to exclude apoA and apoB to avoid multicollinearity issues due to their high correlation with HDL and LDL, respectively ($r_{apoA,HDL} = 0.978$; $P$ value ($P$) $< 2.2 \times 10^{-6}$/$r_{apoB,LDL} = 0.984$; $P < 2.2 \times 10^{-6}$). The minimum $P$ across the remaining lipids was computed, and selection of SNPs was based on those that satisfied genome-wide significance through the minimum $P$ ($P < 5 \times 10^{-8}$). After harmonisation was performed, we clumped the data based on a threshold of $r2 < 0.001$. The main estimation method performed was the IVW, while we additionally implemented the MR-Egger estimate [26] to control for any remaining unmeasured pleiotropy.

## Sensitivity MR analyses

As we observed a positive finding for Lp(a) and PCa outcomes, we performed the following sensitivity analyses in our univariable MR considering only Lp(a) as exposure for total, advanced, and early age onset of PCa.

1. **Sensitivity analysis 1:** As an attempt to increase the statistical power of the univariable MR, we used an eased clumping threshold of $r2 < 0.01$ and refitted the models based on a larger set of IVs.

2. **Sensitivity analysis 2:** Variants that were used as IVs for Lp(a) in the Burgess and colleagues paper [27], based on a clumping threshold of $r2 < 0.4$, were separately fitted to the univariable models to validate findings on a different IV set. Of the 43 IVs used in the paper, 35 IVs were available in both the exposure and outcome dataset. In order to avoid weak instrument bias, we included only 28 genetic variants, which were genome-wide significant for Lp(a). The univariable models were refitted based on these 28 IVs.

3. **Sensitivity analysis 3:** As the *LPA* gene (chromosome 6: 160,531,482–160,664,275) is the main gene associated with Lp(a) concentrations and explains about 70% to 90% of its variability [28], we selected variants located in the *LPA* gene based on a clumping threshold of *r2* < 0.001 to represent strong biological instruments and potentially support the effect of Lp(a). Four such variants were identified and subsequently utilised as IVs.

4. **Sensitivity analysis 4:** Additional robust estimation methods were utilised as part of our sensitivity analyses to control and/or test for horizontal pleiotropy. These included the MR-PRESSO [29] and contamination mixture [30].

As obesity may be considered a probable confounder for lipids and PCa [14], we also performed an additional adjustment for body mass index (BMI) in all the MVMR models, using genetic association data for BMI from UK Biobank [17]. Lp(a) is assembled in the liver [31], whereas liver function/disease has been proposed to influence PCa detection and outcomes [32,33]. Genetic associations for aspartate aminotransferase (AST) and alanine aminotransferase (ALT) were thereby adjusted in a MVMR model including Lp(a) and total PCa. In addition, as kidney disease has been suggested to affect Lp(a) concentrations [34], and creatinine was previously associated with PCa risk [35], we performed another MVMR analysis using Lp(a), creatinine, and total PCa to control for kidney function. All genetic associations for AST, ALT, and creatinine were acquired from UK Biobank [17]. We reviewed the Phenoscanner database [36,37] (*P* threshold = $10^{-5}$) for secondary traits associations of the 10 IVs included in the main univariable analyses for Lp(a) and found 2 that had secondary associations relevant to inflammation and, specifically, aspirin use. We thereby excluded these 2 SNPs from the main univariable Lp(a) analysis on total, advanced, and early age onset PCa. Finally, we performed a post hoc power calculation for our MR analysis [38], where we set the heritability of the exposures to 24% (as reported for Lp(a) by Sinnott-Armstrong and colleagues [18]). Throughout our analyses, we considered significant estimates based on the 95% confidence level. We additionally estimated a Bonferroni and a Holm-Bonferroni corrected *P* for the main univariable analyses on total, advanced, and early age onset PCa, to adjust for the multiple tests performed on each outcome. The total number of tests is reflected upon the number of different lipids we considered for each PCa outcome. Throughout the results section, nominally significant results are reported.

## Analysis plan

Our analysis began by investigating the role of blood lipids in total PCa risk using a univariable MR approach. We used MVMR to adjust for multiple lipid traits and after significance persisted for Lp(a), we performed numerous sensitivity analyses focused primarily on Lp(a) to evaluate the robustness of our finding. We then repeated the same set of analyses on advanced PCa and after observing a similar effect for Lp(a), we decided to specifically test for the effect of Lp(a) on early age onset PCa. Finally, we also performed all univariable analyses for all lipids on early age onset PCa. This study is reported as per the Strengthening the Reporting of Observational Studies in Epidemiology (STROBE) guideline, specific for MR (STROBE-MR) (S1 Checklist) [39].

## Results

Descriptive statistics for lipid measurements were available from UK Biobank [17] and can be seen in S3 Table. Throughout this section, we report results based solely on the IVW and weighted median methods. Results from the additional methods we used, including MR-Egger (S4–S8 Tables), MR-PRESSO and contamination mixture estimates (S4 Table), and MVMR

analyses adjusting for BMI, AST, creatinine, and ALT (S9–S12 Tables), can be found in the supplement and were in general in agreement with the main analyses presented in the text below. Results for the univariable analysis that excludes aspirin-related IVs can be found in S13 Table. The marginal associations of the genetic instruments with exposures, outcomes, and confounders/mediators are shown in S14–S22 Tables. Our power calculation showed that any of the 3 PCa outcomes had a power of 90% or higher to detect an effect of 1.091 or larger (S1 Fig).

## Univariable MR

**Total PCa.**   The univariable MR analysis showed that genetically predicted HDL ($OR_{IVW}$ = 0.994; 95% CI = [0.942,1.051]; $P$ = 0.825), TG ($OR_{IVW}$ = 1.026; 95% CI = [0.961,1.105]; $P$ = 0.449), apoA ($OR_{IVW}$ = 1.025; 95% CI = [0.970,1.083]; $P$ = 0.372), and apoB ($OR_{IVW}$ = 1.026; 95% CI = 0.961,1.094]; $P$ = 0.411) concentrations were not associated with total PCa risk (S5 Table). In contrast, the odds ratio (OR) of total PCa was 1.088 per standard deviation (SD) increase in genetically predicted LDL (95% CI = [1.010,1.162]; $P$ = 0.016). This association was, however, not supported by the weighted median approach (OR = 1.016; 95% CI = [0.942,1.094]; $P$ = 0.669). This raised concerns for potential pleiotropic effects present in our model, and results were further assessed in the multivariable model.

Genetically predicted Lp(a) had an insignificant association on total PCa as estimated from the IVW ($OR_{IVW}$ = 1.066; 95% CI = [0.909,1.249]; $P$ = 0.431) method (Table 1), but the OR of total PCa in the weighted median approach was 1.091 per SD increase in genetically predicted Lp(a) (95% CI = [1.028,1.157]; $P$ = 0.004). Alteration of the clumping threshold in Sensitivity analysis 1 resulted in a higher number of IVs fitted to our model, which supported a significant effect estimate for Lp(a) in both the IVW ($OR_{IVW}$ = 1.076; 95% CI = [1.016,1.114]; $P$ = 0.012) and weighted median approaches ($OR_{weighted\ median}$ = 1.066; 95% CI = [1.012,1.123]; $P$ = 0.016). Sensitivity analysis 2, which included IVs according to the Burgess and colleagues paper [27], also supported a relationship between genetically elevated Lp(a) and total PCa ($OR_{IVW}$ = 1.037; 95% CI = [1.009,1.066]; $P$ = 0.010, $OR_{weighted\ median}$ = 1.044; 95% CI = [1.026,1.061]; $P$ = $6.58 \times 10^{-7}$). Sensitivity analysis 3, which involved variants located in the *LPA* gene only, supported an even stronger OR ($OR_{weighted\ median}$ = 1.439; 95% CI = [1.280,1.619]; $P$ = $1.80 \times 10^{-9}$) for total PCa per SD increase in genetically predicted Lp(a).

**Advanced PCa.**   The univariable MR analysis did not reveal any significant association between blood lipids and advanced PCa risk (HDL; $OR_{IVW}$ = 0.977; 95% CI = [0.905,1.051]; $P$ = 0.552, LDL; $OR_{IVW}$ = 1.067; 95% CI = [0.970,1.74]]; $P$ = 0.191, TG; $OR_{IVW}$ = 1.004; 95% CI = [0.923,1.094]; $P$ = 0.921, Lp(a); $OR_{IVW}$ = 1.064; 95% CI = 0.910,1.245]; $P$ = 0.435, ApoA; $OR_{IVW}$ = 1.001; 95% CI = [0.932,1.073], $P$ = 0.991, ApoB; $OR_{IVW}$ = 0.992; 95% CI = [0.914,1.073], $P$ = 0.837) (Table 1, S6 Table). However, Sensitivity analysis 2 for Lp(a), which included IVs of the Burgess and colleagues paper [27], supported an association between genetically elevated Lp(a) ($OR_{weighted\ median}$ = 1.033; 95% CI = [1.001,1.065]; $P$ = 0.046) and advanced PCa. In addition, Sensitivity analysis 3, which restricted to variants in the *LPA* gene, supported an association between genetically elevated Lp(a) and advanced PCa ($OR_{weighted\ median}$ = 1.388; 95% CI = [1.213,1.590]; $P$ = $2.14 \times 10^{-6}$).

**Early age onset of PCa.**   HDL, apo A, and apo B were not associated with early age onset PCa in any of the methods used (HDL; $OR_{IVW}$ = 0.989; 95% CI = [0.816,1.104]; $P$ = 0.847, Apo A; $OR_{IVW}$ = 1.044; 95% CI = [0.933,1.166]]; $P$ = 0.452, Apo B; $OR_{IVW}$ = 1.125; 95% CI = [0.974,1.094]; $P$ = 0.108). Genetically predicted LDL was associated with early age onset PCa via the IVW method (OR = 1.226; 95% CI = [1.037,1.451]; $P$ = 0.017) but not via the pleiotropy-robust methods, which again raised concerns for potential pleiotropy as with the total

**Table 1. Univariable estimates of genetically predicted Lp(a) on each PCa outcome.**

| | | Method | OR | 95% CI | P | P_bon[†] | P_rank[◆] | P_holm[◆] |
|---|---|---|---|---|---|---|---|---|
| **Total PCa** | **Main Analysis** | IVW | 1.066 | [0.909,1.249] | 0.431 | 1 | 4 | 1 |
| | | **Weighted Median** | **1.091** | **[1.028,1.157]** | **0.004**\*\* | **0.024**\* | **1** | **0.024**\* |
| | **Sensitivity Analysis 1** | **IVW** | **1.076** | **[1.016,1.114]** | **0.012**\* | - | | |
| | | **Weighted Median** | **1.066** | **[1.012,1.123]** | **0.016**\* | - | | |
| | **Sensitivity Analysis 2** | **IVW** | **1.037** | **[1.009,1.066]** | **0.010**\*\* | - | | |
| | | **Weighted Median** | **1.044** | **[1.026,1.061]** | **$6.582 \times 10^{-7}$**\*\*\* | - | | |
| | **Sensitivity Analysis 3** | IVW | 1.228 | [0.960,1.570] | 0.104 | - | | |
| | | **Weighted Median** | **1.439** | **[1.280,1.619]** | **$1.799 \times 10^{-9}$**\*\*\* | - | | |
| **Advanced PCa** | Main Analysis | IVW | 1.064 | [0.910,1.245] | 0.435 | 1 | 2 | 1 |
| | | Weighted Median | 1.071 | [0.973,1.179] | 0.158 | 0.948 | 1 | 0.948 |
| | Sensitivity Analysis 1 | IVW | 1.051 | [0.981,1.127] | 0.158 | - | | |
| | | Weighted Median | 1.051 | [0.974,1.135] | 0.197 | - | | |
| | **Sensitivity Analysis 2** | IVW | 1.024 | [0.992,1.058] | 0.139 | - | | |
| | | **Weighted Median** | **1.033** | **[1.001,1.065]** | **0.046**\* | - | | |
| | **Sensitivity Analysis 3** | IVW | 1.226 | [0.957,1.570] | 0.107 | - | | |
| | | **Weighted Median** | **1.388** | **[1.213,1.590]** | **$2.138 \times 10^{-6}$**\*\*\* | - | | |
| **Early age onset PCa** | **Main Analysis** | IVW | 1.169 | [0.927,1.473] | 0.188 | 1 | 3 | 0.752 |
| | | **Weighted Median** | **1.257** | **[1.107,1.426]** | **$4.000 \times 10^{-4}$**\*\*\* | **$2.4 \times 10^{-3}$**\*\*\* | **1** | **$2.4 \times 10^{-3}$**\*\*\* |
| | **Sensitivity Analysis 1** | **IVW** | **1.215** | **[1.096,1.349]** | **$2.028 \times 10^{-4}$**\*\*\* | - | | |
| | | **Weighted Median** | **1.217** | **[1.084,1.365]** | **0.001**\*\*\* | - | | |
| | **Sensitivity Analysis 2** | **IVW** | **1.076** | **[1.027,1.126]** | **0.002**\*\* | - | | |
| | | **Weighted Median** | **1.079** | **[1.036,1.124]** | **$3.201 \times 10^{-4}$**\*\*\* | - | | |
| | **Sensitivity Analysis 3** | IVW | 1.481 | [0.907,2.418] | 0.116 | - | | |
| | | **Weighted Median** | **1.502** | **[1.276,1.770]** | **$1.038 \times 10^{-6}$**\*\*\* | - | | |

ORs for each PCa outcome are reported per SD increase in genetically predicted Lp(a). The main analysis included IVs based on a clumping threshold of 0.001; Sensitivity analysis 1 is based on an eased clumping threshold of 0.01, Sensitivity analysis 2 is based on a different IV set from another paper, and, finally, Sensitivity analysis 3 is based upon variants located in the LPA gene. Associations of $P < 0.05$ are shown in bold. $P \in (0.05, 0.1]$

\*$P \in (0.01, 0.05]$

\*\*$P \in (0.001, 0.01]$

\*\*\*$P \in (0, 0.001]$.

[†]$P\_bon$ is the Bonferroni corrected $P$, considering the total number of tests performed in the main analysis for total, advanced, and early age onset PCa. This reflects a total of 6 univariable analyses performed on each outcome (one for each lipid).

[◆]$P\_holm$ is the Bonferroni-Holm adjusted $P$, while $P\_rank$ is the rank of the $P$ for Lp(a) compared to other lipids.

IV, instrumental variable; IVW, inverse variance weighting; Lp(a), lipoprotein A; OR, odds ratio; PCa, prostate cancer; SD, standard deviation.

PCa results. In addition, genetically predicted TG was found to be significantly associated with early age onset PCa in the weighted median approach (OR = 1.223; 95% CI = [1.041,1.438]; $P = 0.015$) (S7 Table). Genetically predicted Lp(a) was associated with an increased risk of early age onset PCa in the main univariable analysis ($OR_{weighted\ median} = 1.257$; 95% CI = [1.107,1.426]; $P = 4.00 \times 10^{-4}$) (Table 1). All univariable sensitivity analyses performed confirmed a significant relationship between genetically elevated Lp(a) and early age onset of PCa. [(Sensitivity analysis 1; $OR_{IVW} = 1.215$; 95% CI = [1.096,1.349]; $P = 2.03 \times 10^{-4}$, $OR_{weighted\ median} = 1.217$; 95% CI = [1.084,1.365]; $P = 0.001$), (Sensitivity analysis 2; $OR_{IVW} = 1.076$; 95% CI = [1.027,1.126]; $P = 0.002$, $OR_{weighted\ median} = 1.079$; 95% CI = [1.036,1.124]; $P = 3.20 \times 10^{-4}$), (Sensitivity analysis 3; $OR_{weighted\ median} = 1.502$; 95% CI = [1.276,1.770]; $P = 1.04 \times 10^{-6}$)].

## Multivariable MR

As an attempt to control for pleiotropic pathways that could arise from the relationship between different lipid traits, we incorporated an MVMR model including Lp(a), HDL, LDL, and TG jointly as exposures for each PCa outcome. The significant association observed between genetically predicted LDL and total PCa in the univariable MR attenuated in the MVMR model and was no longer significant (OR = 1.052; 95% CI = [0.973,1.134]; $P$ = 0.183) (Table 2). However, after adjusting for HDL, LDL, and TG, genetically predicted Lp(a) remained significantly and positively associated with total PCa (OR = 1.068; 95% CI = [1.005,1.134]; $P$ = 0.034). Additional adjustment for BMI led to an almost unaltered OR for total PCa risk per SD increase in genetically predicted Lp(a) (OR = 1.066; 95% CI = [1.008,1.129]; $P$ = 0.026) (S8 Table). Genetically predicted Lp(a) was associated at borderline significance with advanced PCa after adjusting for multiple lipid traits (OR = 1.078; 95% CI = [0.999,1.163]; $P$ = 0.055). Additional adjustment for BMI led to an OR of 1.075 (95% CI: [1,1.155]; $P$ = 0.050). The effects of LDL and TG that were previously observed in the univariable MR for early age onset PCa were no longer significant in the MVMR after adjusting for other lipid traits. (LDL; OR = 1.112; 95% CI = [0.948,1.305]; $P$ = 0.192], TG; OR = 1.062; 95% CI = [0.908,1.242]; $P$ = 0.45). However, genetically elevated Lp(a) remained significantly associated with early age onset of PCa (OR = 1.150; 95% CI = [1.015,1.303]; $P$ = 0.028), in agreement with the univariable MR analysis. Adjustment for BMI yielded a similar effect of 1.155 (95% CI = [1.029,1.297]; $P$ = 0.015). IVW estimates for all lipids from the MVMR can be seen in Table 2 below, whereas the IVW BMI-adjusted results can be found in S9 Table. The effect size of Lp(a) did not attenuate after adjusting for other genetic confounders we considered (AST, ALT, and creatinine) (S10–S12 Tables). We compared the multivariable Lp(a) estimates from all the analyses performed on total, advanced, and early age onset PCa with the

**Table 2. MVMR results for each PCa outcome.**

|  | Biomarker | OR | 95% CI | $P$ |
|---|---|---|---|---|
| **Total PCa** | HDL | 1.009 | [0.946,1.077] | 0.775 |
|  | LDL | 1.052 | [0.973,1.134] | 0.183 |
|  | TG | 1.026 | [0.953,1.106] | 0.485 |
|  | **Lp(a)** | **1.068** | **[1.005,1.134]** | **0.034**[*] |
| **Advanced PCa** | HDL | 0.993 | [0.917,1.077] | 0.872 |
|  | LDL | 1.007 | [0.916,1.106] | 0.892 |
|  | TG | 1.003 | [0.914,1.101] | 0.952 |
|  | Lp(a) | 1.078 | [0.999,1.163] | 0.055. |
| **Early age onset PCa** | HDL | 1.024 | [0.894,1.174] | 0.732 |
|  | LDL | 1.112 | [0.948,1.305] | 0.192 |
|  | TG | 1.062 | [0.908,1.242] | 0.450 |
|  | **Lp(a)** | **1.150** | **[1.015,1.303]** | **0.028**[*] |

Each estimate (OR) is based on the multivariable IVW method and represents the direct effect of the risk factor on the respective outcome after controlling for the other 3 biomarkers in MVMR. ORs are reported per SD increase in the respective biomarker. Genetically elevated Lp(a) is significantly associated with total and early age onset PCa, whereas it is associated also at borderline significance with advanced PCa. Associations of $P < 0.05$ are shown in bold. $P \in (0.05,0.1]$

[*]$P \in (0.01,0.05]$

[**]$P \in (0.001,0.01]$

[***]$P \in (0,0.001]$.

HDL, high-density lipoprotein; IVW, inverse variance weighting; LDL, low-density lipoprotein; Lp(a), lipoprotein A; MVMR, multivariable MR; OR, odds ratio; PCa, prostate cancer; TG, triglyceride.

univariable estimates and additional sensitivity analyses through a panel of 3 distinct forest plots (Fig 1). IVs, according to variants in the *LPA* gene (Sensitivity analysis 3), supported the strongest effect between genetically predicted Lp(a) concentrations and each PCa outcome.

## Discussion

Our MR analyses provided evidence that genetically predicted Lp(a) concentration is associated with risk of total, advanced, and early age onset PCa. There was little evidence that any of the other lipids (i.e., LDL, HDL, TG, apoA, and apoB) were associated with PCa outcomes. Specifically, IVs located in the *LPA* gene supported the strongest and most significant Lp(a) associations for total, advanced, and early age onset PCa. Given the strong regulation of Lp(a) levels by the *LPA* gene region [28], the latter findings are based on strong instruments with a clear biological function. Adjustment for multiple lipid traits and BMI in the MVMR models further supported a significant association of genetically predicted Lp(a) on total, advanced, and early age onset PCa.

The null associations observed for genetically predicted HDL on total PCa agree with findings from 2 previous MR analyses [14,15], and the null findings for TG are also supported in the Bull and colleagues [14] paper. In our analysis, there was some evidence for a significant LDL association with total PCa risk, though this was likely a false indication due to pleiotropy, as suggested by the MVMR model, which indicated no association with LDL. As Lp(a) includes an LDL component [40], the attenuation of LDL to the null in the MVMR could be attributed to independent actions of Lp(a) itself, as we did not observe any association between other Lp(a) components and PCa risk. Alternative explanations are that Lp(a) concentrations are less affected by statins compared to LDL [41], thus genetically predicted Lp(a) may be more accurate for current actual levels than genetically predicted LDL, or that the association for Lp(a) dominates over LDL due to the high between-person variability of Lp(a) [18]. The

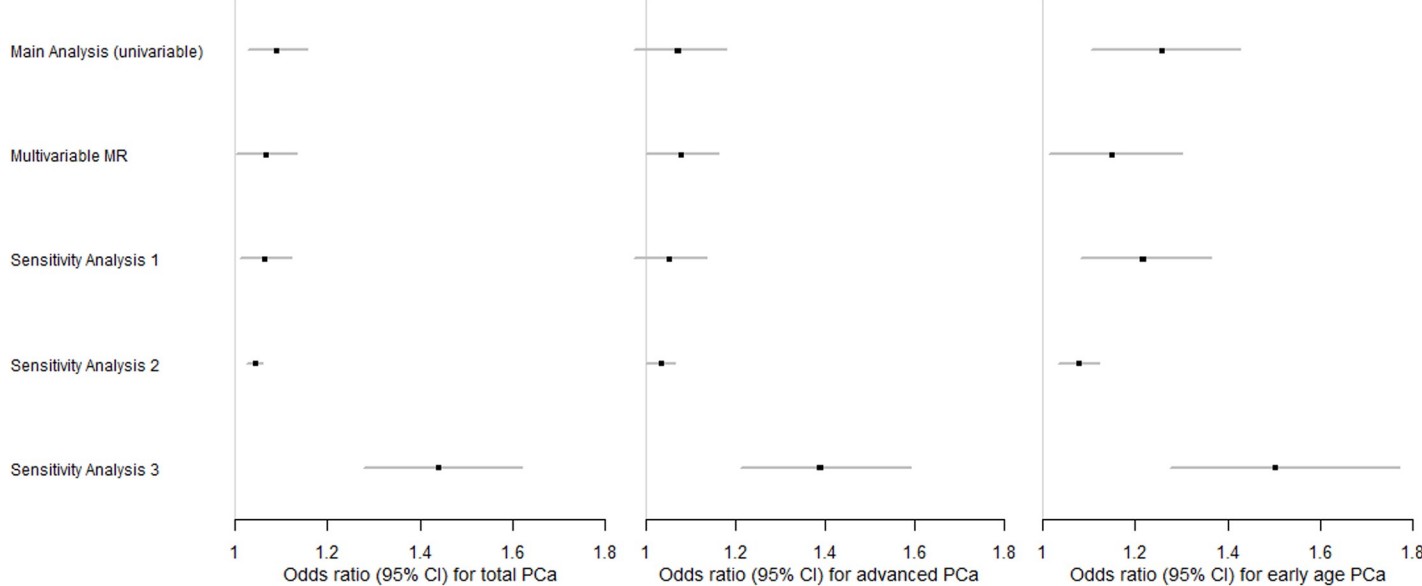

**Fig 1. Forest plots of the Lp(a) effects observed in different analyses based on each PCa type.** The main and sensitivity analyses estimates are based on the weighted median approach, whereas MVMR includes the IVW estimates. Sensitivity analyses 1–3 refer to the univariable models. Sensitivity analysis 1 is based on an eased clumping threshold of 0.01, Sensitivity analysis 2 includes an IV set based on another paper, and, finally, Sensitivity analysis 3 is based upon variants located in the *LPA* gene. Each square represents the OR for each PCa outcome, reported per SD increase in the biomarker, with the 95% CI represented by the error bars. IV, instrumental variable; IVW, inverse variance weighting; Lp(a), lipoprotein A; MVMR, multivariable MR; OR, odds ratio; PCa, prostate cancer; SD, standard deviation.

authors of the Bull and colleagues paper [14] suggested a potential role of LDL and TG in advanced/high-grade PCa; our findings for LDL and TG in advanced PCa risk are not in agreement. Our analyses included adjustment for multiple lipid traits in contrast with the previously mentioned papers, which we believe plays a vital role in MR analysis modelling blood lipids, given the high correlation between them. As far as we are aware, no previous MR study has investigated the role of apoA and apoB in PCa risk. Our null results are nonetheless in agreement with observational studies by Katzke and colleagues [42], which involved the prospective EPIC–Heidelberg cohort and Borgquist and colleagues [43], which was based on the prospective Malmö Diet and Cancer Study (MDCS).

To the best of our knowledge, no previous MR study has examined the role of Lp(a) in PCa risk. The positive association observed for Lp(a) in total PCa was supported by the observational study of Katzke and colleagues [42]. Results showed that top versus bottom quartile levels of Lp(a) were associated with a 47% higher risk of PCa (OR = 1.47; 95% CI = 1.06 to 2.04). Wang and colleagues [44], another observational study, examined the role of Lp(a) in high-risk PCa via a multivariable regression adjusted for age, BMI, hypertension, diabetes, coronary artery disease, and lipid-lowering drugs. They observed that high Lp(a) levels were positively associated ($OR_{Q4 \text{ vs. } Q1}$ = 2.687; 95% CI = 1.113 to 6.491; $P$ = 0.028) with high-risk PCa, which agrees with our findings for advanced PCa in the MVMR analysis. In addition, a recent large prospective cohort among 211,754 men in UK Biobank [45] observed a suggestive positive association between Lp(a) and PCa risk (hazard $ratio_{per SD}$ = 1.02; 95% CI: [0.99,1.06]). Our literature review did not reveal any studies investigating the role of Lp(a) in early age onset PCa.

A range of different biological mechanisms have been proposed to explain pro-cancer effects of cholesterol at the cellular level, including cell proliferation, inflammation, membrane organisation, and steroidogenesis [46]. It is unclear whether total cholesterol or any lipoprotein particle is the causal factor, and the potential pathophysiological mechanisms of Lp(a) have not been well studied. However, emerging evidence from the cardiovascular literature supports pleiotropic functions of Lp(a) and complex mediation pathways with other lipid particles [47]. Lp(a) is highly heritable (heritability = 24%) [18], with the majority of individuals having low Lp(a) levels. However, African Americans, which are known to have the highest risk for PCa, tend to also have higher circulating Lp(a) levels [28]. It has been previously observed that mean Lp(a) concentrations for African Americans are 106 (60 to 180) nmol/l, whereas Caucasians such as non-Hispanic whites have mean Lp(a) concentrations of 24 (7.2 to 79.2) nmol/l [48]. Although the exact explanation behind ethnic discrepancies in PCa is currently unknown, it has been hypothesised that access to healthcare may play a partial role in this. Yet, given that disparities in PCa risk are apparent regardless of cancer detection issues, it is likely that biological factors are key drivers of this phenomenon [49]. Two recent papers have further provided evidence of a different immune response [50] and inflammatory signalling [51] for African Americans versus Caucasians, which can be linked to their poorer PCa prognosis. Considering Lp(a) as a modifier of the immune/inflammatory response [52], the increased Lp(a) concentrations in African Americans and our observed association between genetically elevated Lp(a) and PCa, we hypothesise that Lp(a) may partially account for some of the observed discrepancies in PCa risk by ethnicity. Future large-scale genomic studies in African ancestry populations [53] would be required to evaluate the hypothesis that Lp(a) can explain discrepancies in PCa risk by race.

We note several limitations to our research. There is no direct way to prove that the second and third MR assumptions hold and as such, violations would result in biased MR estimates. A large number of robust methods and sensitivity analyses were used to probe into potential violations mainly due to horizontal pleiotropy, but its presence cannot be excluded. The samples analysed for our main MR analyses were restricted to Europeans to avoid issues with

heterogeneity, which is required for a two-sample MR [54]. However, this may affect generalisability of the results, which are restricted to those of European ancestry. The number of variants associated with Lp(a) was limited in comparison to other lipids. Initially, 5,894 variants were identified to be associated with Lp(a) at GWAS significance, whereas all other lipids had more than 10,000 associated variants. This then resulted in a final sample size of 10 variants due to LD clumping in the main univariable analysis, which may have decreased our statistical power. However, after relaxing the LD clumping threshold in our sensitivity analyses, we included more variants, the findings of which corroborated the main results. In addition, some previous observational studies have suggested potential threshold effects for cholesterol concentrations and PCa [55,56], which cannot be studied in two-sample MR with summary-level data, and future one-sample MR studies are warranted.

Apart from the caveats in our study, there are also several strengths that should be noted. We used an MR study design, in which the outcome of interest is compared between genotypes, analogous to that between treatment and placebo groups in a randomised controlled trial. However, inference should be made with great caution as alterations of genetically predicted risk factors are not identical to those due to a drug or dietary intervention [57]. Secondly, as lipids are dependent on each other for their main functionalities [16], it is important to control for pleiotropic pathways that may arise from these dependencies. One method to do so is via the use of MVMR, which allows to include genetic information on exposures that may correlate with each other into a joint multivariable model [58], and our study forms the first such MVMR conducted to investigate the relationship between various lipid traits and PCa risk. Thirdly, the use of UK Biobank data allowed us to include information on underexamined lipid traits such as Lp(a), apoA, and apoB, in comparison to previous PCa studies, which mainly considered HDL, LDL, and TG. In addition, we have sex-specific genetic associations, and this allowed us to work with male-specific data, which are relevant to PCa. Finally, our analyses are based on large sample sizes, which were acquired from UK Biobank [17] and the PRACTICAL consortium [20].

In summary, findings from this study point towards a positive association between genetically predicted Lp(a) concentrations and risk of total, advanced, and early age onset PCa. Screening for high Lp(a) concentrations could possibly be investigated in the future to identify high-risk groups for PCa. Given that Lp(a) concentrations depend significantly on genetics [59], modification of Lp(a) levels may be achieved by developing Lp(a)-lowering drugs [60] that might be on the horizon. A personalised approach in repurposing lipid drugs that target Lp(a) directly for high-risk individuals could consequently be considered, upon replication of our findings, to study their effectiveness against PCa prevention. The mechanisms behind the observed association remain, however, unclear given the uncertainty underlining the pleiotropic physiological functions of the *LPA* gene itself, which controls about 70% to 90% of the Lp (a) variability [40,59]. Further research into this complex gene such as colocalization analysis would be required to understand more of its functionality and consequently its role in PCa risk.

## Supporting information

**S1 Checklist. STROBE-MR checklist.** Reporting document following the STROBE-MR guidelines for our study.
(DOCX)

**S1 Fig. Post hoc power calculation for all PCa outcomes.** The figure displays the power to detect a significant association on the y-axis against and the true effect size on the x-axis. The different line types indicate the 3 different cases and control numbers for the PCa outcomes.

In red, we highlight the observed effect size by the median MR method for total PCa (OR = 1.091; 95% CI = [1.028,1.157]). This power calculation shows that any of the 3 PCa outcomes had a power of 90% or higher to detect an effect of 1.091 or larger. MR, Mendelian randomisation; OR, odds ratio; PCa, prostate cancer.
(TIF)

**S1 Table. Heritability of each blood lipid as estimated in a GWAS of Sinnott-Armstrong and colleagues [18].** The estimates represent total heritability and the methodology followed was the HESS. The number of SNPs fitted in each model for the main univariable analysis is also reported. GWAS, genome-wide association study; HESS, heritability estimation summary statistics; SNP, single nucleotide polymorphism.
(XLSX)

**S2 Table. GWAS that were meta-analysed for the PCa data as reported in Schumacher and colleagues [20].** Studies 1–7 refer to previous GWAS, whereas the ELLIPSE OncoArray was a custom developed high-density genotyping array. GWAS, genome-wide association study; PCa, prostate cancer.
(XLSX)

**S3 Table. Descriptive statistics of each blood lipid.** Lp(a) is positively skewed, whereas the rest of the lipids are approximately normally distributed. Measurements are based on all samples (both sexes) in UK Biobank [17]. Lp(a), lipoprotein A.
(XLSX)

**S4 Table. MR-Egger, MR-Presso, and Contamination Mixture estimates for Lp(a) on each PCa outcome from the univariable analysis.** The contamination mixture method may indicate 2 distinct CIs associated with a single estimate. Associations of $P < 0.05$ are shown in bold. $P \in (0.05,0.1]$, $^*P \in (0.01,0.05]$, $^{**}P \in (0.001,0.01]$, $^{***}P \in (0,0.001]$. Lp(a), lipoprotein A; MR, Mendelian randomisation; PCa, prostate cancer.
(XLSX)

**S5 Table. Univariable MR estimates for each lipid on total PCa.** Genetically elevated LDL is significantly associated with total PCa only through the IVW approach, whereas the pleiotropy-robust methods do not support this association. Associations of $P < 0.05$ are shown in bold. $P \in (0.05,0.1]$, $^*P \in (0.01,0.05]$, $^{**}P \in (0.001,0.01]$, $^{***}P \in (0,0.001]$. IVW, inverse variance weighting; LDL, low-density lipoprotein; MR, Mendelian randomisation; PCa, prostate cancer.
(XLSX)

**S6 Table. Univariable MR estimates for each lipid on advanced PCa.** None of these lipids are associated with advanced PCa. Associations of $P < 0.05$ are shown in bold. $P \in (0.05,0.1]$, $^*P \in (0.01,0.05]$, $^{**}P \in (0.001,0.01]$, $^{***}P \in (0,0.001]$. MR, Mendelian randomisation; PCa, prostate cancer.
(XLSX)

**S7 Table. Univariable MR estimates for each lipid on early age onset PCa.** Genetically elevated LDL is significantly associated with early age onset PCa only through the IVW approach, whereas the pleiotropy-robust methods do not support this association. In addition, the weighted median approach supports a significant association for TG. Associations of $P < 0.05$ are shown in bold. $P \in (0.05,0.1]$, $^*P \in (0.01,0.05]$, $^{**}P \in (0.001,0.01]$, $^{***}P \in (0,0.001]$. IVW, inverse variance weighting; LDL, low-density lipoprotein; MR, Mendelian randomisation;

PCa, prostate cancer; TG, triglyceride.
(XLSX)

**S8 Table. MR-Egger estimates for Lp(a) and each PCa outcome.** Sensitivity analyses refer to the univariable MR and are as follows: Sensitivity analysis 1 includes variants according to an eased clumping threshold of r2 < 0.01, Sensitivity analysis 2 includes SNPs for Lp(a) according to a different IV set, whereas Sensitivity analysis 3 is based on variants included in the LPA gene. Associations of $P < 0.05$ are shown in bold. $P \in (0.05,0.1]$, $^*P \in (0.01,0.05]$, $^{**}P \in (0.001,0.01]$, $^{***}P \in (0,0.001]$. IV, instrumental variable; Lp(a), lipoprotein A; MR, Mendelian randomisation; PCa, prostate cancer; SNP, single nucleotide polymorphism.
(XLSX)

**S9 Table. IVW estimates from the MVMR model adjusted for BMI.** Genetically elevated Lp(a) is associated with overall, advanced, and early onset of PCa in these models. Associations of $P < 0.05$ are shown in bold. $P \in (0.05,0.1]$, $^*P \in (0.01,0.05]$, $^{**}P \in (0.001,0.01]$, $^{***}P \in (0,0.001]$. BMI, body mass index; IVW, inverse variance weighting; Lp(a), lipoprotein A; MVMR, multivariable MR; PCa, prostate cancer.
(XLSX)

**S10 Table. IVW and MR-Egger estimates from the total PCa MVMR model adjusted for Lp(a) and AST.** Genetically elevated Lp(a) is significantly associated with total PCa through the IVW method. Associations of $P < 0.05$ are shown in bold. $P \in (0.05,0.1]$, $^*P \in (0.01,0.05]$, $^{**}P \in (0.001,0.01]$, $^{***}P \in (0,0.001]$. AST, aspartate aminotransferase; IVW, inverse variance weighting; Lp(a), lipoprotein A; MR, Mendelian randomisation; MVMR, multivariable MR; PCa, prostate cancer.
(XLSX)

**S11 Table. IVW and MR-Egger estimates from the total PCa MVMR model adjusted for Lp(a) and creatinine.** Genetically elevated Lp(a) is significantly associated with total PCa through the IVW method and associated at borderline significance with total PCa with the MR-Egger approach. Associations of $P < 0.05$ are shown in bold. $P \in (0.05,0.1]$, $^*P \in (0.01,0.05]$, $^{**}P \in (0.001,0.01]$, $^{***}P \in (0,0.001]$. IVW, inverse variance weighting; Lp(a), lipoprotein A; MR, Mendelian randomisation; MVMR, multivariable MR; PCa, prostate cancer.
(XLSX)

**S12 Table. IVW and MR-Egger estimates from the total PCa MVMR model adjusted for Lp(a) and ALT.** Genetically elevated Lp(a) is significantly associated with total PCa through the MR-Egger method. Associations of $P < 0.05$ are shown in bold. $P \in (0.05,0.1]$, $^*P \in (0.01,0.05]$, $^{**}P \in (0.001,0.01]$, $^{***}P \in (0,0.001]$. ALT, alanine aminotransferase; IVW, inverse variance weighting; Lp(a), lipoprotein A; MR, Mendelian randomisation; MVMR, multivariable MR; PCa, prostate cancer.
(XLSX)

**S13 Table. Univariable analysis on each PCa and Lp(a) after removing aspirin-associated SNPs Associations of $P < 0.05$ are shown in bold.** $P \in (0.05,0.1]$, $^*P \in (0.01,0.05]$, $^{**}P \in (0.001,0.01]$, $^{***}P \in (0,0.001]$. Lp(a), lipoprotein A; PCa, prostate cancer; SNP, single nucleotide polymorphism.
(XLSX)

**S14 Table. Association between SNPs and each lipid biomarker and BMI.** The following list includes IVs used in the main analysis, MVMR (adjusted for HDL, LDL, TG, Lp(a), and BMI), and Sensitivity analyses 1–4. BMI, body mass index; HDL, high-density lipoprotein; IV,

instrumental variable; LDL, low-density lipoprotein; Lp(a), lipoprotein A; MVMR, multivariable MR; SNP, single nucleotide polymorphism; TG, triglyceride.
(XLSX)

**S15 Table. Association between SNPs and total PCa.** The following list includes IVs used in the main analysis, MVMR (adjusted for HDL, LDL, TG, Lp(a), and BMI), and Sensitivity analyses 1–4. BMI, body mass index; HDL, high-density lipoprotein; IV, instrumental variable; LDL, low-density lipoprotein; Lp(a), lipoprotein A; MVMR, multivariable MR; PCa, prostate cancer; SNP, single nucleotide polymorphism; TG, triglyceride.
(XLSX)

**S16 Table. Association between SNPs and advanced PCa.** The following list includes IVs used in the main analysis, MVMR (adjusted for HDL, LDL, TG, Lp(a), and BMI), and Sensitivity analyses 1–4. BMI, body mass index; HDL, high-density lipoprotein; IV, instrumental variable; LDL, low-density lipoprotein; Lp(a), lipoprotein A; MVMR, multivariable MR; PCa, prostate cancer; SNP, single nucleotide polymorphism; TG, triglyceride.
(XLSX)

**S17 Table. Association between SNPs and early age onset PCa.** The following SNPs include IVs used in the main analysis, MVMR (adjusted for HDL, LDL, TG, Lp(A), and BMI), and Sensitivity analyses 1–4. BMI, body mass index; HDL, high-density lipoprotein; IV, instrumental variable; LDL, low-density lipoprotein; Lp(a), lipoprotein A; MVMR, multivariable MR; PCa, prostate cancer; SNP, single nucleotide polymorphism; TG, triglyceride.
(XLSX)

**S18 Table. Association between SNPs, Lp(a), AST, and total PCa.** The following SNPs include IVs used in the additional MVMR analysis on Lp(a), AST, and total PCa only. AST, aspartate aminotransferase; IV, instrumental variable; Lp(a), lipoprotein A; MVMR, multivariable MR; PCa, prostate cancer; SNP, single nucleotide polymorphism.
(XLSX)

**S19 Table. Association between SNPs, Lp(a), creatinine, and total PCa.** The following SNPs include IVs used in the additional MVMR analysis on Lp(a), creatinine, and total PCa only. IV, instrumental variable; Lp(a), lipoprotein A; MVMR, multivariable MR; PCa, prostate cancer; SNP, single nucleotide polymorphism.
(XLSX)

**S20 Table. Association between SNPs, Lp(a), ALT, and total PCa.** The following SNPs include IVs used in the additional MVMR analysis on Lp(a), ALT, and total PCa only. ALT, alanine aminotransferase; IV, instrumental variable; Lp(a), lipoprotein A; MVMR, multivariable MR; PCa, prostate cancer; SNP, single nucleotide polymorphism.
(XLSX)

**S21 Table. Association between SNPs, Lp(a), and total PCa.** The following SNPs include IVs used in the main univariable analysis on Lp(a) and total PCa only. The overlap between this analysis and Sensitivity analysis 3 (SNPs in the LPA gene region) is highlighted in yellow. IV, instrumental variable; Lp(a), lipoprotein A; PCa, prostate cancer; SNP, single nucleotide polymorphism.
(XLSX)

**S22 Table. Association between SNPs included in the LPA gene (Sensitivity analysis 3), Lp(a), and total PCa.** Lp(a), lipoprotein A; PCa, prostate cancer; SNP, single nucleotide

polymorphism.
(XLSX)

**S1 File. Members from the PRACTICAL Consortium, CRUK, BPC3, CAPS, and PEGA-SUS.**
(PDF)

## Acknowledgments

Members from the PRACTICAL Consortium, CRUK, BPC3, CAPS, and PEGASUS:

Rosalind A. Eeles, Christopher A. Haiman, Zsofia Kote-Jarai, Fredrick R. Schumacher, Sara Benlloch, Ali Amin Al Olama, Kenneth R. Muir, Sonja I. Berndt, David V. Conti, Fredrik Wiklund, Stephen Chanock, Ying Wang, Catherine M. Tangen, Jyotsna Batra, Judith A. Clements, APCB BioResource (Australian Prostate Cancer BioResource), Henrik Grönberg, Nora Pashayan, Johanna Schleutker, Demetrius Albanes, Stephanie Weinstein, Alicja Wolk, Catharine M. L. West, Lorelei A. Mucci, Géraldine Cancel-Tassin, Stella Koutros, Karina Dalsgaard Sørensen, Eli Marie Grindedal, David E. Neal, Freddie C. Hamdy, Jenny L. Donovan, Ruth C. Travis, Robert J. Hamilton, Sue Ann Ingles, Barry S. Rosenstein, Yong-Jie Lu, Graham G. Giles, Robert J. MacInnis, Adam S. Kibel, Ana Vega, Manolis Kogevinas, Kathryn L. Penney, Jong Y. Park, Janet L. Stanford, Cezary Cybulski, Børge G. Nordestgaard, Sune F. Nielsen, Hermann Brenner, Christiane Maier, Jeri Kim, Esther M. John, Manuel R. Teixeira, Susan L. Neuhausen, Kim De Ruyck, Azad Razack, Lisa F. Newcomb, Davor Lessel, Radka Kaneva, Nawaid Usmani, Frank Claessens, Paul A. Townsend, Jose Esteban Castelao, Monique J. Roobol, Florence Menegaux, Kay-Tee Khaw, Lisa Cannon-Albright, Hardev Pandha, Stephen N. Thibodeau, David J. Hunter, Peter Kraft, William J. Blot, Elio Riboli.

## Author Contributions

**Conceptualization:** Anna Ioannidou, Konstantinos K. Tsilidis, Verena Zuber.

**Data curation:** Anna Ioannidou.

**Formal analysis:** Anna Ioannidou.

**Investigation:** Anna Ioannidou, Eleanor L. Watts, Aurora Perez-Cornago, Elizabeth A. Platz, Ian G. Mills, Timothy J. Key, Ruth C. Travis, Verena Zuber.

**Methodology:** Anna Ioannidou, Verena Zuber.

**Project administration:** Konstantinos K. Tsilidis, Verena Zuber.

**Software:** Anna Ioannidou.

**Supervision:** Konstantinos K. Tsilidis, Verena Zuber.

**Writing – original draft:** Anna Ioannidou, Konstantinos K. Tsilidis, Verena Zuber.

**Writing – review & editing:** Anna Ioannidou, Eleanor L. Watts, Aurora Perez-Cornago, Elizabeth A. Platz, Ian G. Mills, Timothy J. Key, Ruth C. Travis, Konstantinos K. Tsilidis, Verena Zuber.

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
