## [Editor Report · Decision Letter 0]

9 Jun 2021

Dear Dr Zuber, 

Thank you for submitting your manuscript entitled "The relationship between Lipoprotein A and other lipids with prostate cancer risk: A multivariable Mendelian randomisation study." for consideration by PLOS Medicine.

Your manuscript has now been evaluated by the PLOS Medicine editorial staff as well as by an academic editor with relevant expertise and I am writing to let you know that we would like to send your submission out for external peer review.

Please re-submit your manuscript within two working days, i.e. by Jun 11 2021 11:59PM.

Kind regards,

Callam Davidson

Associate Editor

PLOS Medicine

---

## [Decision Letter · Decision Letter 1]

24 Aug 2021

Dear Dr. Zuber,

Thank you very much for submitting your manuscript "The relationship between Lipoprotein A and other lipids with prostate cancer risk: A multivariable Mendelian randomisation study." (PMEDICINE-D-21-02496R1) for consideration at PLOS Medicine. 

Your paper was evaluated by an associate editor and discussed among all the editors here. It was also discussed with an academic editor with relevant expertise, and sent to independent reviewers, including a statistical reviewer. The reviews are appended at the bottom of this email and any accompanying reviewer attachments can be seen via the link below:

[LINK]

In light of these reviews, we will not be able to accept the manuscript for publication in the journal in its current form, but we would like to consider a revised version that addresses the reviewers' and editors' comments. We cannot make any decision about publication until we have seen the revised manuscript and your response, and we plan to seek re-review by one or more of the reviewers. 

We hope to receive your revised manuscript by Sep 14 2021 11:59PM. Please email us (plosmedicine@plos.org) if you have any questions or concerns.

We look forward to receiving your revised manuscript. 

Sincerely,

Callam Davidson, 

Associate Editor

PLOS Medicine

plosmedicine.org

Please add this statement to the manuscript's Competing Interests: "VZ is a paid statistical consultant on PLOS Medicine's statistical board."

In your data availability statement, please provide contact information for data requests (web or email address) from the Practical Consortium. Note that a study author cannot be the contact person for the data.

Please remove the ‘Funding’ statement from the title page, in the event of publication this information will be published as metadata.

Please open the Discussion with a short, clear summary of the article's findings, before moving on to discussing findings within the context of existing research.

Please ensure that the study is reported according to the STROBE guideline, and include the completed STROBE checklist as Supporting Information. Please add the following statement, or similar, to the Methods: "This study is reported as per the Strengthening the Reporting of Observational Studies in Epidemiology (STROBE) guideline (S1 Checklist)."

Did your study have a prospective protocol or analysis plan? Please state this (either way) early in the Methods section.

The paper does not discuss the involvement of PSA levels in relation to the findings. PSA levels of >100ng/mL were used as a criteria for inclusion - if possible, could the authors please comment on how high/low PSA levels affected the analysis or the sensitivity of the screen?

Comments from the reviewers:

Reviewer #1: Ioannidou et al., reported a multivariable Mendelian randomization study on the relationship between blood lipids with prostate cancer risk. They investigated the LDL, HDL, TG, apoA/B, lipoprotein A and GWAS from UK Biobank and PRACTICAL consortium for ~167K individuals. They identified only Lp(a) instead of other lipids was consistently associated with higher total prostate cancer risk in different MR models, the result remained significantly when restricted samples in advanced or early age onset prostate cancer. The paper was well-written, and analysis was carefully performed with extensive sensitivity analysis. See below, some of my questions.

1/ for the study design with case-control setting, it could more informative if a description of control selection can be addressed. How did they match, by age, bmi etc or by propensity score? With current setting of sample size, does it provide enough power to detect significant signals?

2/ with only inclusion of European ancestry in the study, it needs further clarification how to decide whether the participant is EA, what is the cut-off. Should the genetic admixture be included in the MR analysis? Authors mentioned the disparity of the prostate cancer risk with African American has the highest risk. Does the UK biobank have sufficient data from African American to carry the similar MR as the current one?

3/ IVs selection, it seems there are several criteria of r-square, 0.001 for main analysis, and 0.01 and even 0.4 for sensitivity analysis. Why did authors choose those cut-offs? How much inflations of SNP number was achieve due to such cut-offs? It is necessary to discuss the pros and cons on relaxing the cut-off to 0.4 to include more IVs, as it could increase the power but also with so many features, it could be potentially overfitted the model. Could combination of machine learning with MR better help with IVs selection?

4/ Could authors provide a table of IVs and their statistics to be involved for the main analysis for each lipid, especially Lp(a), do those IVs fall into the LPA gene regions, overlapped with sensitivity analysis 3? 

5/ for the multivariable MR, can author explore the effect of aspirin use as adjustment to the analysis, as many studies showed aspirin decreases lipids level and advanced prostate cancer risk.

Reviewer #2: The paper uses Mendelian randomization to examine the effects of six blood lipid traits on the risk of prostate cancer. Using data from the UK Biobank and the PRACTICAL consortium, the authors implement both univariate and multivariable MR and identify a link between LP(a) and prostate cancer, but little evidence of a causal effects for the other blood lipid traits.

The paper is well written and makes proper use of the relevant statistical methodology. Its conclusions prove to be robust to a number of sensitivity analyses, and the discussion puts the association between LP(a) and prostate cancer into perspective, comparing it with the relevant literature. While reading the manuscript I only had very minor comments, which I will not bother the authors with. Instead, I recommend that the paper is accepted for publication.

Reviewer #3: This is a well written manuscript focusing on the relationship between blood lipids and prostate cancer risk. Two-sample MR and MVMR analyses illustrated the causality of Lp(a) and increased PCa risk. The authors adopted proper methods and carefully presented their results, and the limitations and advantages were discussed and well addressed. Nevertheless, I have some concerns for authors to respond.

1. Page 6, first paragraph: The authors run the univariable analysis on all blood lipids for both total and advance Pca, but for early age onset Pca they focused the univariable analysis on Lp(a), why only for Lp(a) and what are the findings for other lipids?

2. Page 6, last paragraph: Burgess et al identified 43 IVs in their paper [reference 27], but only 28 of them were used in the current manuscript, will bias be induced on the basis of the selective 28 IVs?

3. Page 7, 3rd paragraph: obesity was used as a confounding factor for lipids and Pca, but lipids including Lp(a) may not be independent of obesity, and this may violate the 2nd assumption of using MR, how do the authors explain this potential bias?

4. ALT and AST are the established key biomarkers for liver function. Why did the author only select AST in MVMR analysis? What are the impact from ALT? Are there any conflicting findings with ALT? I would suggest the authors to provide relevant results with ALT to enhance their results.

5. Page 12, last paragraph: I did not find a description for the main findings with regards to Fig 1.

6. There are several format errors or typos, such as, capital letters in title, line 232 a redundant "=", etc. Please check the entire manuscript carefully.

7. Please unify the format of p and p-value throughout the manuscript.

Reviewer #4: This study investigates the effects of different lipids on prostate cancer risk using a multivariable Mendelian randomization design. The MR design avoids issues such as unobserved confunding and reverse-causation which are common in observational studies by utilizing predictive genes of different lipids as instrumental variables. The genes are assumed to be randomly allocated to people and thus are independent of both confounders (observed and unobserved) and outcomes (PCa). Therefore, these IVs (i.e. the genes) can help tease out the causal effect of different lipids on PCa. 

Overall, I think this is a well designed and carefully conducted study. The author carefully cleaned the data and conducted various sensitivity analyses to make sure the results are robust. Below are specific comments:

1. The authors claim that this study does not require ethics statement. Given the individual-level data used in this study, I am not sure whether that's the case.

2. The authors mention that there are restrictions of data availability

3. Line 104-105: What is inferred_sex? What is the model and what these covariates are adjusted for? I think you need to describe the GWA data a bit more for reader to understand what these data are and how those data are used in this study. 

4. Line 108: I am not sure what is "heritability estimation from summary statistics". May need a bit explanation for readers who are not familiar with GWA data. 

5. Line 116: what are these imputation methods? Are you imputing missing values or what?

6. Line 135: How the p-value of 5*10-8e is determined?

7. Line 136: What do you mean by removing inconsistencies. What exactly did you do? It could be clearer.

8. Line 141-142: Why did you focus the UA on Lp(a). Did you pre-plan this or did you peek the data to make such decision? 

9. Line 182-184: Bonferroni-correction is known to be very conservative. Did you consider some other corrections which may be more efficient, such as Holm, Benjamini-Hochberg, and Hommel correction, etc.

10. Line 267, Table 2: Did you do any correction for the p-values of the MVMR for multiple testing? The p-values are not very small so I am worried that after correcting for multiple testing, the p-values may not be significant anymore.

[LINK]

---

## [Decision Letter · Decision Letter 2]

27 Oct 2021

Dear Dr. Zuber,

Thank you very much for re-submitting your manuscript "The relationship between lipoprotein A and other lipids with prostate cancer risk: A multivariable Mendelian randomisation study." (PMEDICINE-D-21-02496R2) for review by PLOS Medicine.

I have discussed the paper with my colleagues and the academic editor and it was also seen again by three reviewers. I am pleased to say that provided the remaining editorial and production issues are dealt with we are planning to accept the paper for publication in the journal.

[LINK]

We look forward to receiving the revised manuscript by Nov 03 2021 11:59PM.   

Sincerely,

Callam Davidson, 

Associate Editor 

PLOS Medicine

plosmedicine.org

Requests from Editors:

The STROBE-MR statement was recently published (https://jamanetwork.com/journals/jama/fullarticle/2785494), please ensure the most up-to-date reference is cited.

Line 54: Please update to 'were associated'.

Line 83: Please update to 'could play a potentially important role'

Please carefully check your references as I noted several minor inconsistencies (e.g. references 5, 22, 39)

Please add '[preprint]' to references 19 and 58.

Comments from Reviewers:

Reviewer #1: Authors have fully addressed my questions.

Reviewer #3: I am satisfied with authors' responses, and do not have further comments.

Reviewer #4: I think the authors have properly addressed the comments/questions raised by the reviewers. I am happy with the revised version and think this paper is ready for publication. Congratulations!

[LINK]

---

## [Editor Report · Decision Letter 3]

3 Nov 2021

Dear Dr Zuber, 

On behalf of my colleagues and the Academic Editor, Dr Ricky Johnstone, I am pleased to inform you that we have agreed to publish your manuscript "The relationship between lipoprotein A and other lipids with prostate cancer risk: A multivariable Mendelian randomisation study." (PMEDICINE-D-21-02496R3) in PLOS Medicine.

PUBLICATION SCHEDULE

Given our busy publication schedule for the remainder of 2021, we are planning to publish your paper in early January 2022 (the exact date will be communicated to you once confirmed).

PRESS

Sincerely, 

Callam Davidson 

Associate Editor 

PLOS Medicine